# Impact of DC Transient Disturbances on Harmonic Performance of Voltage Transformers for AC Railway Applications

**DOI:** 10.3390/s22062270

**Published:** 2022-03-15

**Authors:** Palma Sara Letizia, Davide Signorino, Gabriella Crotti

**Affiliations:** Istituto Nazionale di RIcerca Metrologica, INRIM, 10135 Torino, Italy; d.signorino@inrim.it (D.S.); g.crotti@inrim.it (G.C.)

**Keywords:** instrument transformers, harmonic measurements, DC transient disturbances, power quality in AC railway systems

## Abstract

This paper analyzes the impact of typical DC transient events occurring in railway grids on the frequency performance of instrument transformers (ITs) installed onboard trains and in AC substations for power quality (PQ) applications. PQ monitoring in railway systems is an issue of great interest because it plays a key role in the improvement of energy efficiency. The measurement chain for the PQ measurements, at 15 kV at 16.7 Hz and 25 kV at 50/60 Hz, commonly includes ITs to scale the voltage to levels fitting the input of the measurement units. Nevertheless, the behavior of ITs in the presence of PQ phenomena represents an open issue from a normative point of view, even for those installed in conventional AC power supply systems. In this context, the paper presents a possible definition of DC transient disturbances test waveforms, a measurement procedure, and a setup to assess the impact of these disturbances on the harmonic performances of ITs for railway systems. Preliminary experimental tests carried out on two commercial ITs under wide ranges of variation for the amplitude and the time duration of DC disturbances show that, in some cases, the error introduced in harmonic measurements can exceed 100%.

## 1. Introduction

The energy efficiency improvement in the railway system, promoted by the European Union requires accurate knowledge of the real-time power quality (PQ) by awarding the good quality of power delivered and absorbed. A relevant international standard [1] defines limit variations for frequency and supply voltage, but the monitoring of PQ phenomena is only required at commissioning or in response to problems. The knowledge of the actual PQ level in the European railway system can provide input to the standardization bodies concerning the definition of narrower PQ limits. This would simplify the design of electrical locomotives, reduce their weight, and increase the lifetime of all electrical traction subsystems.

In the scientific literature, the PQ subject has been covered for more than ten years: many types of different events have been recorded and catalogued, and many measurement systems and algorithms devoted to the classification of these phenomena have been developed [2,3,4,5,6,7,8,9,10].

The instrumentation used for PQ monitoring is designed to work with low voltage and low current levels. For this reason, the PQ monitoring at typical voltage levels of 15 kV at 16.7 Hz and 25 kV at 50/60 Hz requires suitable transducers and commonly instrument transformers (ITs) to reduce voltage and current to levels that fit the input of PQ measuring units. These PQ measurement architectures are very similar to those adopted for the PQ monitoring of conventional AC power supply systems operating at Medium Voltage (MV). In the context of the distribution and transmission systems, the PQ topic is deeply discussed in the literature as well as standards. In particular, proper standards define the voltage characteristics and limits of electricity supplied by public distribution networks [11] and methods for the measurement and detection of PQ disturbances [12,13]. However, these limits and algorithms are not directly applicable to the railway world as the phenomena are different, the rolling stocks are in motion, and the time dynamic is different. There is a normative gap regarding IT performance verification in both supply systems when ITs are used in PQ measurements. Information and data on this issue are partially provided from both the literature [14,15,16,17,18,19] and the technical report [20]. Moreover, in this respect, a European metrology research project, EMPIR 19NRM05 IT4PQ [21], has been funded. The main goal of this project is to establish the methods and procedures for assessing the errors introduced by ITs when they are involved in PQ applications.

One of the most studied and observed PQ phenomena in the railway system is the harmonic voltage [22,23]. Harmonic injection is essentially due to the high-speed/high-capacity trains pulse width modulation (PWM) induction motors and converters that are included in uninterruptible power supply (UPS) or installed onboard the trains [24,25].

A typical PQ event of the railway system is the electric arc [26,27,28,29,30,31,32,33]. This phenomenon can occur due to bad contact between the catenary and the pantograph, such as the discontinuity of the ground, supply line wear, a ruined pantograph, etc. From the electrical point of view, in an AC power supply system, the electric arc results in deformations of the voltage and current waveforms that introduce DC transient disturbances.

Another peculiar problem in the AC railway system is the presence of the in-rush current [34]. The in-rush current, for a transformer, is the maximum instantaneous value that the current assumes when it is switched on. This current may be several times the normal full-load current and can last for a few cycles of the input waveform. Since the power supply voltage is supplied single-phase, there are sections of the line, known as neutral sections, in which the train is not powered and proceeds by inertia before passing in a section of the line powered by another phase. Every time this supply change happens, the in-rush events occur, and various transients producing DC components are expected.

DC voltage and more in general, low-frequency disturbances can badly influence the behavior of voltage transformers (VTs) [15]. When a transformer is supplied with a DC waveform, a DC magnetization bias of the magnetic circuit will result. The magnetizing current becomes asymmetrical and presents harmonic components. This harmonic is directly proportional to the ability of the DC to magnetize the core and on the core design. Therefore, the effects strongly depend on the DC disturbance’s amplitude and time duration, the typology of the iron-core, and, in general, the transformer design [35].

In the analysis of PQ events, it is important to distinguish between the events that occur on the railway supply grid and the errors induced by the VTs. In this context, this paper aimed to quantify and analyze the effect of DC transient disturbances occurring in the AC railway system on the performance of inductive VTs involved in PQ monitoring. As the first step, a mathematical description of typical transient DC disturbances has been determined. Furthermore, the possible ranges of variation of the disturbance in terms of the maximum amplitude of the DC component and duration have been identified. A suitable experimental setup and a measurement procedure have been developed to evaluate the performance of VT in harmonic measurements both with and without the transient DC disturbance. The measurement procedure and setup have been used to test two MV commercial inductive VT: the first is designed as measurement unit on-board train, at 16.7 Hz at 15 kV and 50 Hz at 25 kV applications (VT${}_{A}$); the second is developed with the same technology as those used in a substation (VT${}_{B}$).

The paper is organized as follows: Section 2 presents the description of the measurement setup adopted to assess the performances of devices with proper test waveforms. In Section 3, the measurement procedure in terms of performed test waveforms is described. In Section 4, a wide range of experimental results obtained on the two VTs under different operating conditions is illustrated. Finally, in Section 5, the discussion of the results and the conclusion are provided.

## 2. Generation and Measurement Setup

The generation and measurement system for the VT characterization at MV is shown in Figure 1, where the functional diagram (a) and a laboratory picture (b) are reported. The MV test signals are obtained using an arbitrary waveform generator (AWG) coupled with a high-voltage power amplifier. The AWG is the National Instrument (NI) PCI Extension for Instrumentation (PXI) 5421 board. The amplifier used is a Trek high-voltage power amplifier. The AWG is housed in a PXI chassis, and the 10 MHz PXI clock is used as a reference clock for its PLL circuitry. A second NI PXI AWG is used to generate a clock signal with a frequency equal to 12.8 MHz, which is provided to the acquisition system as a time base clock. The applied voltage is measured by a reference resistive, capacitive voltage divider (RCVD) designed, built, and characterized at INRIM [36]. The RCVD rated primary voltage is ±30 kV, and its frequency response is flat from DC to 9 kHz. The acquisition system is an NI compact Data Acquisition system (cDAQ) chassis with various boards. The main characteristics of the instrumentation adopted are provided in Table 1.

The overall uncertainty (coverage factor k = 2) of the measurement setup is 70 μV/V and 70 μrad at power frequency, and it increases to 200 μV/V and 350 μrad at 9 kHz for the ratio error and the phase error measurements.

## 3. Measurement Procedure

This section presents the measurement procedure carried out to quantify the impact of DC disturbances on the performances of the VTs. More particularly, the preliminary frequency characterization test and DC disturbance tests are described.

### 3.1. FH1 Test: VT Preliminary Characterization

The VTs are preliminarily characterized with bi-tone test waveforms (in the following FH1, Fundamental plus one Harmonic). The test voltage is composed of a fundamental tone at a rated amplitude and frequency with a superimposed harmonic with amplitude equal to 5% of the rated one and harmonic order *h* in the range from the 2nd to the 200th. The frequency performances of the VTs under test are evaluated in terms of harmonic ratio Equation (Equation 1) and phase errors Equation (Equation 2), according to the following equations:(1)$$\epsilon_{h}=\frac{k_{r}V_{s,h}-V_{p,h}}{V_{p,h}}$$
(2)$$\phi_{h}=\phi_{s,h}-\phi_{p,h}$$
where:$k_{r}=V_{p,r}/V_{s,r}$ is the rated transformation ratio ($V_{p,r}$ and $V_{s,r}$ are the rated primary and secondary voltages);$V_{p,h}$ and $V_{s,h}$ are the root mean square (rms) values of the primary and secondary *h*-order harmonic voltages;$\phi_{p,h}$ and $\phi_{s,h}$ are the phase angles of the primary and secondary *h*-order harmonic voltages.

### 3.2. DC Disturbance Test

The VTs are supplied with test waveforms composed of a stationary and a transient. The stationary signal is a multitone signal composed of the fundamental tone at a rated amplitude and frequency with the first $N_{h}$ harmonic tones superimposed. The transient signal is a decreasing exponential described as in the Equation (Equation 3):(3)$$A\cdot e^{-\frac{t}{\tau }}$$
where:*A* is the amplitude;*t* is the time variable;τ is the time constant.

Therefore, the VTs test waveform can be mathematically expressed as follows:(4)$$\left[A_{f}\cdot sin\left(2\pi f_{f}t\right)+\sum_{h=2}^{N_{h}}\left(A_{h}\cdot sin\left(2\pi hf_{f}t\right)\right)\right]\cdot \left[A\cdot e^{-\frac{t-t_{0}}{\tau }}\cdot u\left(t-t_{0}\right)\right]$$
where:$t_{0}$ is the time instant in which the DC disturbance is applied;$u(t-t_{0})$ is the Heaviside step function;$A_{f}$ is the fundamental amplitude;$f_{f}$ is the fundamental frequency;$A_{h}$ is the harmonic amplitude.

To assess the performances of the VT under different DC disturbances, the test parameters are varied according to the values given in Table 2.

The $\phi_{\mathrm{start}}$ parameter represents the phase of the fundamental tone at $t=t_{0}$. The considered harmonics are the first seven ($N_{h}$ = 7) and their amplitude $A_{h}$ is set to 1% of $A_{f}$.

## 4. Experimental Results

This section presents the results of the tests performed on two different VTs. The analyzed VTs are two commercial VTs for MV phase to ground metering applications. In particular, the VT${}_{A}$ is designed for rolling stock installation and can be used for two different supply systems, whereas VT${}_{B}$ is a transformer that technologically and constructively resembles those installed in railway substations. The VTs main features are summarized in Table 3.

In Section 4.1, the errors introduced by VT${}_{A}$ and VT${}_{B}$ in harmonic measurement without the DC disturbance influence are provided. In Section 4.2, the DC disturbance effects on VT${}_{A}$ and VT${}_{B}$ harmonic performance is quantified.

In all the tests, the detection and measurement of harmonics are performed according to the prescriptions given by the relevant standard [37] which defines the measurement methods and time windows. In [37], the fast Fourier transform (FFT) is the recommended processing tool, and the basic measurement time interval is set to 10 cycles for 50 Hz power systems. However, no standards provide similar indications for power systems operating at 16.7 Hz; for this reason, the authors decided to also consider time windows of 10-cycles for 16.7 Hz.

### 4.1. Reference Tests: FH1

In this subsection, the results of VT characterization under FH1 (see Section 3.1) tests are reported. As can be observed in Figure 2, both the VTs have a quite flat frequency response at first harmonics (within ±1% up to 800 Hz for VT${}_{A}$ and 1900 Hz for VT${}_{B}$). The VT${}_{A}$ has the first resonance frequency at 3250 Hz, that is at harmonic orders *h* = 65 and *h* = 194 if fundamental frequencies $f_{f}$ equal to 50 Hz and 16.7 Hz are considered, respectively. VT${}_{B}$ has the first resonance frequency at 6250 Hz, that is, the 125th harmonic for the 50 Hz fundamental frequency.

The [38,39] standards do not provide limits to the accuracy of VTs used for harmonics or, more generally, PQ measurements. However, indications can be found in [40] for low power instrument transformers (LPITs). This standard extends the accuracy class for the ITs used for quality metering applications. In particular, for 0.5 class LPITs, the standard prescribes that their errors from 100 Hz to 1 kHz (excluded) must not exceed 5%, whereas from 1 kHz (included) to 3 kHz, their errors must not be greater than 10%. Considering this indication, the VT${}_{A}$ complies with the 5% limit up to 1700 Hz, whereas it reaches the 10% limit at 2100 Hz, which means harmonic order h=42 for 50 Hz power frequency and harmonic h=125 for 16.7 Hz. On the contrary, for VT${}_{B}$, the error is below 5% up to 3250 Hz and below 10% up to 4200 Hz.

### 4.2. Impact of DC Disturbance on Harmonic Measurements

This subsection shows the results of transient tests (see Section 3.2) performed on VT${}_{A}$ and VT${}_{B}$.

The generated disturbance at the VT primary side produces a DC transient component superimposed on the multitone signal, as highlighted in Figure 3. Before the disturbance occurs, the errors $\epsilon_{h}$ have time constant values equal to those obtained under the FH1 characterization. When the DC transient disturbance arises, the $\epsilon_{h}$ instantaneously increases and then follows a decreasing dynamic that depends on the characteristic of the generated DC disturbance. This phenomenon is due to the spurious harmonic component at the VT secondary side generated by the iron core non-linearity. After a certain number of τ, the $\epsilon_{h}$ errors return to the value measured under the FH1 test [35]. To quantify the effect of the DC disturbance, the maximum absolute deviation of $\epsilon_{h}$ caused by the transient disturbance is evaluated with respect to the unperturbed condition.

The VT${}_{A}$ and VT${}_{B}$ behaviors under various test conditions are summarized from Figure 4, Figure 5 and Figure 6 where the maximum absolute deviations of the harmonics ratio errors are shown. As a general comment analyzing the single *h* harmonic, it can be observed that the most affected harmonics are the first ones, in particular from the 2nd to the 4th.

Figure 4 provides the results related to the VT${}_{A}$ and VT${}_{B}$ performance in the measurement of the first seven harmonics under disturbances with different amplitudes *A* and time constant τ. It can be observed that the maximum absolute deviations of $\epsilon_{h}$ increase when the parameters *A* and τ increase. In particular, considering the VT${}_{A}$ supplied at 50 Hz under a disturbance with τ=0.05 s (Figure 4 top-left corner), it can be noticed that for A=0.1%, the $\epsilon_{2}$ has a value of 0.03% (comparable with the measurement uncertainty): this value increases to 0.08% with A=1% and to 0.23% with A=2%. When VT${}_{A}$ is supplied at 15 kV at 16.7 Hz, the $\epsilon_{2}$ maximum deviations assume values equal to 0.15%, 0.40% and 0.6% for τ=0.05 s and A=0.1%, 1% and 2%, respectively. Similar results can be observed for VT${}_{B}$; in fact, considering τ=0.05 s (Figure 4 top-right corner), the $\epsilon_{2}$ maximum deviations are equal to 0.10%, 0.43% and 0.70% for A=0.1%, 1% and 2%, respectively. As regards the τ effect, looking at the first column of Figure 4 and only considering the case of A=2%, it can be observed that VT${}_{A}$ at 50 Hz shows $\epsilon_{2}$ maximum variations equal to 0.23%, 0.8% and 106% for τ=0.05 s, 0.2 s and 0.5 s, respectively. Therefore, the increasing of τ of one order of magnitude leads the $\epsilon_{2}$ deviation increasing by a factor 460. The increasing of τ leads to an increase in the $\epsilon_{h}$ deviation of VT${}_{A}$ also when the fundamental amplitude and frequency are set as equal to 15 kV and 16.7 Hz. In this case, the measured $\epsilon_{2}$ maximum variations are 0.15%, 3.73% and 45.6% for τ=0.05 s, 0.2 s and 0.5 s, respectively. The same considerations also apply to VT${}_{B}$, as one can see from the third column of Figure 4 and only considering the *A* = 2% case, it can be observed that $\epsilon_{2}$ maximum deviations take values of 0.70% 3.37% and 46.53% for τ= 0.05 s, 0.2 s and 0.5 s, respectively.

Figure 5 shows a comparison among the maximum error deviation from the 2nd to the 7th harmonic when the DC disturbance occurs at three different time instants: when the phase of the fundamental tone is 0°, 45° and 90°. For the sake of brevity, only the case with A=2%τ = 0.05 s for VT${}_{B}$ is reported. For the device under test, the results show that the variations between the test with $\phi_{\mathrm{start}}$ = 0° and $\phi_{\mathrm{start}}$ = 45° for the 2nd harmonic is approximately 0.12% and between $\phi_{\mathrm{start}}$ = 45° and $\phi_{\mathrm{start}}$ = 90° is 500 ppm, values that are a fraction of the accuracy class of VT${}_{B}$. The values of $\epsilon_{h}$ for the other harmonics basically overlap; thus, in conclusion, it is reasonable to consider the effect of $\phi_{\mathrm{start}}$ as negligible.

The results presented from Figure 4 and Figure 5 refer to VTs supplied with fundamental components $A_{f}$ at a rated voltage. In Figure 6, two other values of fundamental tone amplitudes are considered: 80% and 120% of the rated primary voltage.

As can be observed, for all the harmonic orders, the $\epsilon_{h}$ maximum deviation increases with the fundamental tone amplitude. Considering the A=1% amplitude case, when the VT${}_{A}$ is supplied at 80% of the rated value, the DC disturbance leads to a $\epsilon_{2}$ equal to 2.3%. In contrast, the same quantity reaches 12.30% when the fundamental tone amplitude increases to 120% of the rated one. A considerable increase is observed for the ratio error of the fourth harmonics; in fact, the $\epsilon_{4}$ maximum deviation goes from 0.37% to 11.28% when the primary voltage goes from 80% to 120%. Similar results are observed for VT${}_{B}$: for this VT, the $\epsilon_{2}$ maximum deviation at 80% is 2.40% whereas it is 9.0% at 120%, which is 3.75 times higher. It is important to underline that, according to [38], the VT full operating range from 80% to 120% has to be taken into account in the analysis of the impact of the DC transient since the maximum error increment can be up to one order of magnitude greater.

A summary of the experimental results is provided in Table 4.

## 5. Conclusions

Power quality in AC railway systems is a topic partially covered in the literature and not addressed by the standards; therefore, it represents an open issue. In this scenario, this paper investigated the impact of DC transient disturbances, commonly found in railway systems, on the performances of inductive ITs necessary for PQ monitoring both on trains and in substations. Several tests on a VT for rolling stock and a VT for fixed installation were performed by varying the amplitude and duration of the DC disturbance, the instant in which the transient occurs, and the VT operating voltage. The results show that for events lasting a few milliseconds, even for significant amplitude changes, the impact of the event is negligible compared to the rated performance of the VTs. Different results are obtained for longer durations: with τ = 0.5 s, the second harmonic ratio error for VT${}_{A}$ at 50 Hz increases up to 106%. Experimental results show that the instant and therefore the phase of the signal in which the perturbation occurs can be considered negligible in this analysis. On the contrary, it is important to consider the amplitude of the operating voltage for the VT operating point from 80% to 120% of the nominal amplitude. The maximum error increments can be up to one order of magnitude greater. 

## Figures and Tables

**Figure 1 sensors-22-02270-f001:**
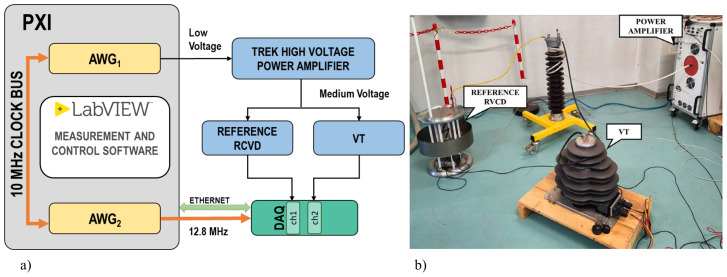
Functional diagram (**a**) and laboratory picture (**b**) of the generation and measurement setup.

**Figure 2 sensors-22-02270-f002:**
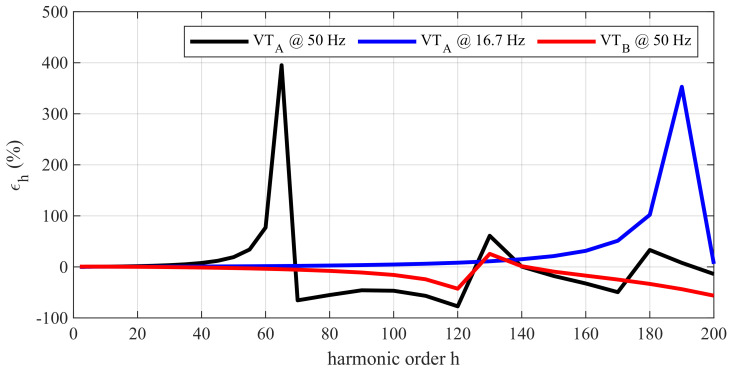
Harmonic ratio error measured under the FH1 test performed on VT${}_{A}$ at 50 Hz, VT${}_{A}$ at 16.7 Hz, and VT${}_{B}$ at 50 Hz.

**Figure 3 sensors-22-02270-f003:**
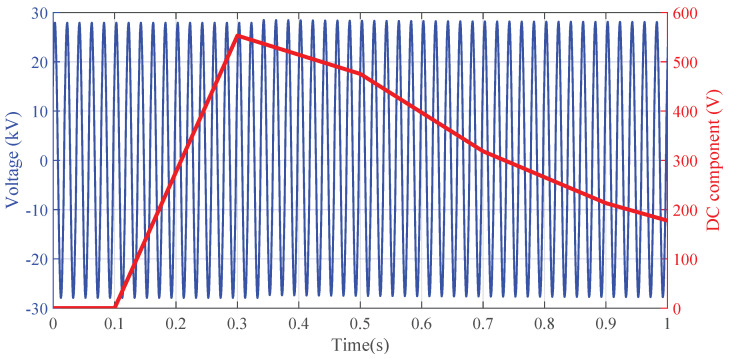
Example of a generated DC transient disturbance with A=2% and τ=0.5 s.

**Figure 4 sensors-22-02270-f004:**
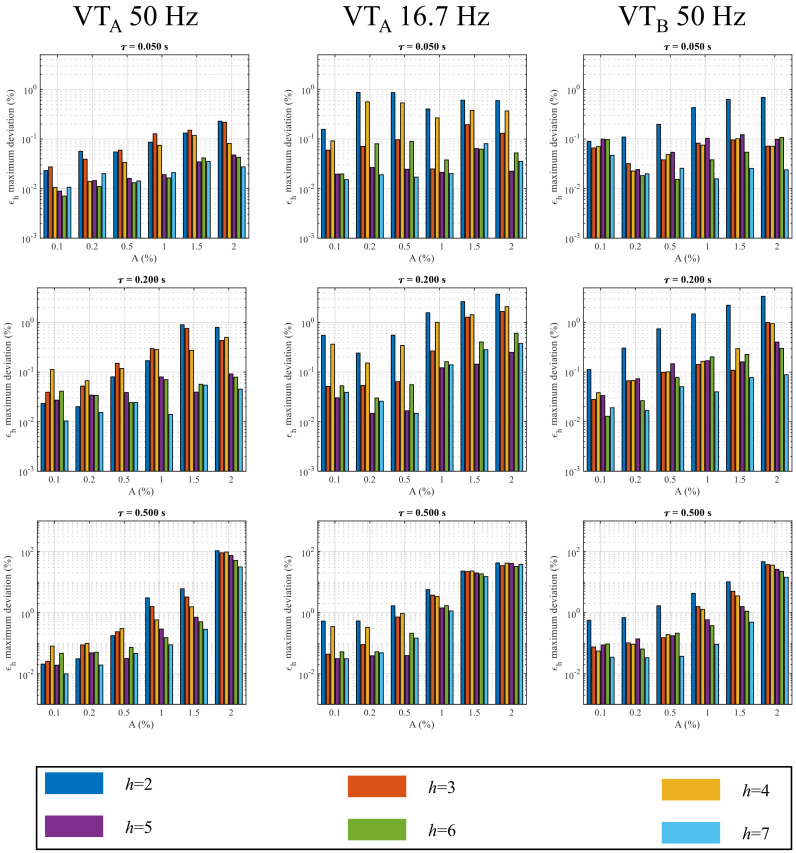
Maximum absolute deviations of harmonics ratio errors for VT${}_{A}$ @ 50 Hz, VT${}_{A}$ @ 16.7 Hz and VT${}_{B}$ @ 16.7 Hz at different values of τ and *A*.

**Figure 5 sensors-22-02270-f005:**
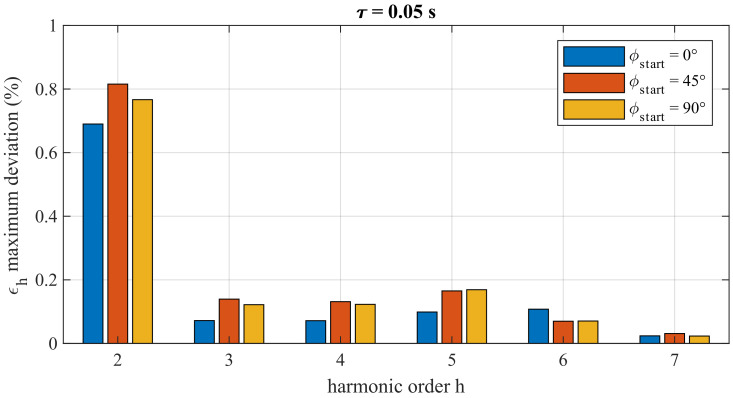
VT${}_{B}$: comparison among the maximum error deviation from the 2nd to the 7th harmonic when the DC disturbance of A=2% and τ=0.05 s occurs at $\phi_{\mathrm{start}}$ = 0°, $\phi_{\mathrm{start}}$ = 45° and $\phi_{\mathrm{start}}$ = 90°.

**Figure 6 sensors-22-02270-f006:**
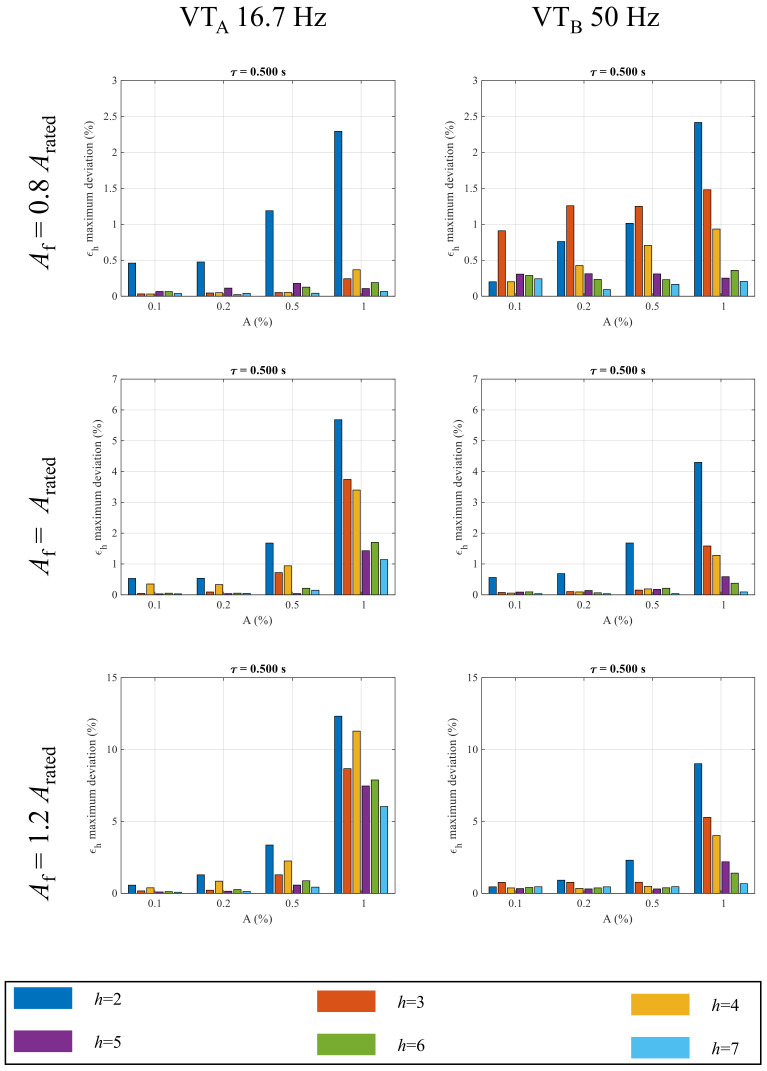
Comparison among the maximum error deviation from the 2nd to the 7th harmonic when the DC disturbance occurs at different fundamental amplitudes.

**Table 1 sensors-22-02270-t001:** Description of the experimental setup instrumentation.

Instrument Type	Name	Main Characteristics
AWG	NI 5421	16-bit, ±12 V, 100 MHz maximum sampling rate
Power amplifier	TREK 30/20A	±30 kV, ±20 mA, DC to 20 kHz bandwidth
Reference sensor	Resistive capacitive voltage divider	±30 kV, DC to 9 kHz bandwidth
Data acquisition	cDAQ NI 9225 and NI 9239	24-bit, 50 kHz sampling rate, ±425 V and ±10 V

**Table 2 sensors-22-02270-t002:** Range of variation of the parameters involved in the tests.

$A_{f}$	*A*	τ	$\phi_{\mathrm{start}}$
(% of Rated)	(% of Fundamental)	(s)	(deg)
80 to 100	0.1 to 2	0.02 to 0.5	0 to 90

**Table 3 sensors-22-02270-t003:** Main features of VTs under test.

Name	Frequency	Primary Voltage	Secondary Voltage	Rated Burden	Application	Accuracy Class and Standard
VT${}_{A}$	50 Hz	25 kV	100 V	25 VA	Rolling stock	0.5
	16.7 Hz	15 kV	60 V		(outdoor)	IEC 60044-7
VT${}_{B}$	50 Hz	$20/\sqrt{3}$ kV	$100/\sqrt{3}$ V	30 VA	MV metering (indoor)	0.5 IEC 61869-3

**Table 4 sensors-22-02270-t004:** Summary of the results obtained from the experimental tests.

Quantity	Impact	Relevance
*h*	High	Most affected harmonics: *h* = 2, *h* = 3, *h* = 4
A	Medium	τ = 0.05 s, ∀ A, ∀ VT, $\epsilon_{h}$ max dev ≤ 1% τ = 0.2 s, A ≥ 1%, VT${}_{A}$ at 16.7 Hz and VT${}_{B}$, $\epsilon_{h}$ max dev ≥ 1% τ = 0.5 s, A ≥ 2%, ∀ VT, $\epsilon_{h}$ max dev ≥ 40%
τ	High	A = 2% τ = 0.05 s, ∀ VT, $\epsilon_{h}$ max dev ≤ 1% τ = 0.2 s, ∀ VT, $\epsilon_{h}$ max dev ≥ 1% τ = 0.5 s, ∀ VT, $\epsilon_{h}$ max dev ≥ 40%
$\phi_{\mathrm{start}}$	Very low	$\epsilon_{h}$ max dev ≤ 0.2%
A${}_{f}$	High	*h* = 2, A = 1%, τ = 0.5 s VT${}_{A}$: max dev 2.3% at 0.8 A${}_{\mathrm{rated}}$; max dev 12.5% at 1.2 A${}_{\mathrm{rated}}$ VT${}_{B}$: max dev 2.5% at 0.8 A${}_{\mathrm{rated}}$; max dev 8.5% at 1.2 A${}_{\mathrm{rated}}$

## Data Availability

Not applicable.

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
