# Peer review of "Impact of DC Transient Disturbances on Harmonic Performance of Voltage Transformers for AC Railway Applications"

_sensors, 2022, doi:10.3390/s22062270_

Round 1
Reviewer 1 Report
It is known that Power Quality (PQ) can be measured using an instrument transformer (IT) up to harmonic order 40. Above harmonic order 40, generally speaking, results cannot be accepted without a detailed check of the IT ratio at higher frequencies.
Can you give some new conclusion regarding PQ measurements using an IT at frequencies higher than 40 harmonics?
How can we use your results for some other ITs?
Author Response
It is known that Power Quality (PQ) can be measured using an instrument transformer (IT) up to harmonic order 40. Above harmonic order 40, generally speaking, results cannot be accepted without a detailed check of the IT ratio at higher frequencies.
Can you give some new conclusions regarding PQ measurements using an IT at frequencies higher than 40 harmonics?
Ans: The reviewer is right; the accuracy of the IT beyond the 40th harmonic must be verified. Commonly, VTs with primary voltage in the order of 20 kV present the first resonance for frequencies in the [2-3] kHz range [1]. This means that the error introduced by such VTs in the measurement of harmonics from the 40th order can exceed 100%.
The frequency accuracy of the two analyzed VTs has been evaluated up to the 200th harmonic (Fig 2). Detailed comments about their accuracy for high-order harmonics have been added.
[1]: M. Klatt, J. Meyer, M. Elst and P. Schegner, "Frequency Responses of MV voltage transformers in the range of 50 Hz to 10 kHz," Proceedings of 14th International Conference on Harmonics and Quality of Power - ICHQP 2010, 2010, pp. 1-6, doi: 10.1109/ICHQP.2010.5625484.
How can we use your results for some other ITs?
Ans: The authors thank the reviewer for this question. The effect of the DC component is strictly due to the design of the transformer and the sizing of the magnetic core. For this reason, similar results can be observed for VTs operating at the same induction and magnetic field points as those analyzed in the paper. On the contrary, for VTs operating in different conditions, experimental tests are required. A sentence that clarifies this concept has been added in the paper.

Reviewer 2 Report
This paper analyses the impact of DC transient events on the frequency performance of ITs. The methods, analysis and results are adequate. I have some minor comments below: 1) I would add more literature on the topic. The present background lacks fault identification methods, the sensitivity of devices used in AC power quality applications, etc. 2) At the end of the introduction, add a paragraph explaining why your research article is unique and your main contributions. 3) Please add the accuracy rate of the results shown in section 4. I would imagine that the LabVIEW application would give a typical ± range of the measurements. 4) I suggest by the end of the results section, you summarise the results by outlining/offering a baseline of the increase for the VTA for the different tested harmonics, so other researchers can quickly populate your conclusions.
Author Response
This paper analyses the impact of DC transient events on the frequency performance of ITs. The methods, analysis, and results are adequate.
I have some minor comments below:
- I would add more literature on the topic. The present background lacks fault identification methods, the sensitivity of devices used in AC power quality applications, etc.
Ans: The introduction section has been modified to improve the literature review.
- At the end of the introduction, add a paragraph explaining why your research article is unique and your main contributions.
Ans: The introduction section has been extended describing more in detail the main contributions and innovations of the paper
3) Please add the accuracy rate of the results shown in section 4. I would imagine that the LabVIEW application would give a typical ± range of the measurements.
Ans: The accuracy of the results has been specified in Section 2.
- I suggest by the end of the results section, you summarise the results by outlining/offering a baseline of the increase for the VTA for the different tested harmonics, so other researchers can quickly populate your conclusions.
Ans: The authors thank and agree with the reviewer’s suggestion. A new table (Table 4) has been added at the end of the experimental results section to provide a summary of the main findings.

Reviewer 3 Report
The paper subject is technically interesting. However, the following concerns should be responded to carefully:
- The paper’s title should be revised. It seems the railway should be added to the paper’s title.
- The structure of the abstract should be revised. Firstly, the motivations of this research should be addressed in the abstract. Afterward, the research gaps should be clarified. Then, the contributions should be highlighted. Finally, the procedures and specifications of the test system and case studies can be discussed.
- Keywords should be revised. Instead of PQ, the full expression should be used.
- The literature review should be improved.
- The experimental tests and achievements should be highlighted.
- The specifications of experimental setups could be listed in one Table.
- The main contributions should be addressed clearly at the end of the introduction.
Author Response
The paper subject is technically interesting. However, the following concerns should be responded to carefully:
- The paper’s title should be revised. It seems the railway should be added to the paper’s title.
Ans: The paper title is changed in accordance with the reviewer’s suggestion
- The structure of the abstract should be revised. Firstly, the motivations of this research should be addressed in the abstract. Afterwards, the research gaps should be clarified. Then, the contributions should be highlighted. Finally, the procedures and specifications of the test system and case studies can be discussed.
Ans: Thanks for your advice. The structure of the abstract has been revised according to the reviewer’s suggestions
- Keywords should be revised. Instead of PQ, the full expression should be used.
Ans: Keywords have been modified
- The literature review should be improved.
Ans: The introduction section has been modified to improve the literature review.
- The experimental tests and achievements should be highlighted.
Ans: Text has been modified adding a new Table (Table 4) to highlight the results obtained by experimental tests and achievements reached thanks to this work.
- The specifications of experimental setups could be listed in one Table.
Ans: Thanks for this suggestion. A new Table (Table 1) that resumes the capabilities of the experimental setup has been introduced.
- The main contributions should be addressed clearly at the end of the introduction.
Ans: The introduction section has been extended describing more in detail the main contributions and innovations of the paper.

Reviewer 4 Report
The paper based on an experimental work that investigates DC transients effects on the accuracy of VTs. DC transient signal with different magnitudes and time constants are superimposed fundamental frequency voltage signal and performance of two different VTs are compared.
It is known that superimposed dc will saturate the core of the VTs. Accordingly, measurement should be influenced by this saturation. Main parameters for the saturation is the magnitudes of the superimposed DC components. However the depicted results in the paper suggest that time constant of the dc components significantly distort the measurements.
Could authors elaborate their experimental settings in terms of the measurement duration and time? The measurement system may provide some kind of average value through a time window. This kind of averaging may curtail the died away dc components effects.
There are also discrepancy between figures 4 and 6. Since Af is Arated in figure 6 middle cases, the results should be the same as figure 4 bottom results. But they are not compatible. Also figure 6 caption is a bit complicated needs better wording.
Instrument transformers are produced with different classes. Could authors give some indication about transformers class under test? Also, again instrument transformers are produced for measurement and protection devices. Measurement transformers are designed to saturate to protect measurement equipment in excessive voltages and currents. On the other hand protection transformers are designed to be immune for the saturation. Could authors comment on the type of the instrument transformer as well.
Author Response
The paper based on an experimental work that investigates DC transients effects on the accuracy of VTs. DC transient signal with different magnitudes and time constants are superimposed fundamental frequency voltage signal and performance of two different VTs are compared.
It is known that superimposed dc will saturate the core of the VTs. Accordingly, measurement should be influenced by this saturation.
Ans: The reviewer is right; the presence of the DC component at the VT primary side can lead the iron core to saturation. However, the authors have chosen DC transient component amplitudes and durations to be low enough to avoid the saturation of the VTs during the experimental tests.
Main parameters for the saturation is the magnitudes of the superimposed DC components. However the depicted results in the paper suggest that time constant of the dc components significantly distort the measurements.
Ans: The reviewer’s consideration is correct. The two main parameters of the DC transient disturbances are the amplitude and the time constant. From experimental tests, it is observed that the time constant has a more significant impact on harmonic performance. This result is explained considering that higher τ implies that the DC components decrease slowly and assume high values for longer.
Could authors elaborate their experimental settings in terms of the measurement duration and time? The measurement system may provide some kind of average value through a time window. This kind of averaging may curtail the died away dc components effects.
Ans: The consideration of the reviewer is correct. The time window affects the measurement results when transient events are considered. In the case of the phenomenon presented in the article, applying the measurement algorithm over smaller time intervals provides higher values of harmonic amplitudes, whereas elaboration performed over larger intervals provides lower ones.
However, new sentences have been added in section 4 to clarify that all the measurements have been performed using a fixed time window equal to 10-cycles of the fundamental frequency, according to the basic time interval defined by the International Standard IEC 61000-4-7.
There are also discrepancy between figures 4 and 6. Since Af is Arated in figure 6 middle cases, the results should be the same as figure 4 bottom results. But they are not compatible. Also figure 6 caption is a bit complicated needs better wording.
Ans: The authors thank the reviewer for highlighting the mistake. We did not repeat the same test of Af=Arated, but the wrong figure was selected. Moreover, caption of Fig. 6 has been modified.
Instrument transformers are produced with different classes. Could authors give some indication about transformers class under test? Also, again instrument transformers are produced for measurement and protection devices. Measurement transformers are designed to saturate to protect measurement equipment in excessive voltages and currents. On the other hand protection transformers are designed to be immune for the saturation. Could authors comment on the type of the instrument transformer as well.
Ans: Thanks for the suggestions, in Table 3 the accuracy classes of VTs have been added. Moreover, a sentence clarifying the typology (metering) of VT has been introduced.

Round 2
Reviewer 3 Report
There is no further comments. The author have tried to respond to all review comments.
Reviewer 4 Report
Authors revised the manuscript according to reviewer comments. The paper can be published as is.